# Premature Ejaculation Symptoms Are Associated with Sexual Excitability: Empirical Support for the Hyperarousability Model

**Daniel Ventus** *  **and Patrick Jern**

Department of Psychology, Åbo Akademi University, 20500 Turku, Finland; pjern@abo.fi
* Correspondence: dventus@abo.fi

**Abstract:** Premature ejaculation (PE) is a common sexual complaint among men, but its etiology is poorly understood. Previous studies on the dual control model of sexuality has revealed that propensities for sexual excitation and inhibition can contribute to sexual dysfunctions, but few studies have included a measure of premature ejaculation. We sought to explore whether PE is associated with sexual excitation or inhibition. We applied structural equation models to data from a large population-based sample of Finnish adult men. The analyses supported a four-factor solution for the sexual inhibition/sexual excitation short-form scale. The clearest result was that increased symptoms of PE were associated with a greater propensity for sexual excitation ($\beta = 151$, $p < 001$, $n = 2953$). Importantly, this excitation was intrapersonal, as opposed to stemming from social activities. The results imply that men with PE may have stronger and more rapid reactions to sexual stimuli, which in turn may lead to ejaculating earlier than desired.

**Keywords:** premature ejaculation; early ejaculation; rapid ejaculation; sexual excitation; sexual inhibition; dual control model

## 1. Introduction

A vast body of evidence suggests that premature ejaculation (PE) is one of the most common sexual complaints presented by men, with 20–30% of men reporting recurring ejaculations before they wish to do so during partnered sexual activity [1]. Historically, PE has been hypothesized to be a "psychological" disorder (where anxiety, and especially performance anxiety, has been hypothesized to be the key causal explanation for PE symptoms), whereas more recent research has tended to focus on neurobiological and genetic causes [2]. While considerable efforts have taken place, especially over the last three decades, to understand the etiology of PE symptoms, the etiology of PE remains poorly understood.

Much of recent research appears to support a theoretical framework to understand PE etiology centered around sexual excitability. Simply put, this model proposes that excitation can increase over time until an ejaculatory threshold is reached, and once it is crossed, the ejaculatory reflex can no longer be inhibited (e.g., [3]). The model is derived from the dual control model of sexual response [4], according to which the level of sexual arousal depends on the adaptive interplay between two competing neurobiological factors, namely sexual excitation and sexual inhibition. There is interindividual variation in the propensity for sexual excitation and inhibition, and the extremes of the distributions of variance in these traits are hypothesized to be associated with an increased risk of developing sexual problems [5]. The most common way to measure sexual excitation and inhibition is the Sexual Inhibition/Sexual Excitation Scales [6]. This measure further divides sexual inhibition into two separate factors, where the first (SIS1) is hypothesized to measure sexual inhibition due to performance failure, whereas the other (SIS2) is thought to measure inhibition due to the threat of consequences of sexual performance [4].

In general, it does indeed seem to be the case that the dual control model is basically valid: several studies have found evidence for an association between sexual dysfunctions and either high sexual inhibition or low sexual excitation [7,8]. However, in the case of PE, the most reasonable hypothesis would perhaps be that its symptoms should be associated with high rather than low sexual excitation [3]. This notion is indirectly supported by the fact that desensitizing-based treatment protocols for PE appear to be effective on a group level [9]. It does not appear, however, as straightforward to formulate directed hypotheses for a possible association between sexual inhibition and PE. It is plausible that PE could be a consequence of lacking inhibition (e.g., if a man does not hold himself back during sexual activity, it may result in rapid ejaculation), but it seems equally (if not more) plausible that sexual inhibition could rather be a consequence of PE (e.g., someone starts avoiding sex due to fear of performance failure). These seemingly contradictory hypotheses could plausibly hold true simultaneously in different subgroups.

Very few studies have empirically tested these associations. To our knowledge, the first of these was a survey conducted by Bancroft, Carnes, Janssen, and Long [10], in which no association between self-reported rapid ejaculation and measures of sexual excitation and inhibition were found. However, the aforementioned study did not employ a validated measure of PE, making it difficult to discern the number of participants with clinically significant symptoms (some of the co-authors of that study pointed out in a later review that "only 15 men presented with PE as their only problem" [4]). In a more recent study, Nowosielski, Kurpisz, Kowalczyk, and Lew-Starowicz [11] found, using a convenience sample of 498 men, a small but significant positive correlation between lifetime prevalence of PE and sexual inhibition due to performance failure. In summary, previous research concerning possible associations between sexual excitation and inhibition and PE symptoms has generated conflicting results, and may not be representative of the broader population.

In the present study, we sought to use a large, population-based sample to elucidate possible associations between sexual excitation/sexual inhibition. Based on previous literature, we explored the following hypotheses: (1) PE and sexual excitation are positively associated, (2) PE and sexual inhibition are positively associated, and (3) PE and sexual inhibition are negatively associated.

## 2. Materials and Methods

The present study was conducted as part of a larger population-based data collection involving twins and siblings of twins identified through the Central Population Registry in Finland (for a more detailed description of the sample, see, e.g., [12]). The registry is a computerized national register containing basic information about all Finnish and foreign citizens residing in Finland on a permanent or temporary basis [13]. Eligibility criteria were (a) having Finnish as first language, (b) residency in Finland at the time of gathering addresses, and (c) age of at least 18. A total of 33,211 letters were sent by mail, inviting recipients to respond to an online survey using an individual eight-character code for identification purposes. Participants were offered to enter a raffle that contained 40 gift vouchers to a network of companies operating in retail and services, worth €100 each. Two reminder letters were sent 2 to 3 weeks apart to participants who had not responded. In total, we received 9564 responses, giving a response rate of 29%. Consent to participate in the study was provided by 9319 respondents (97%). Of these participants, 2992 were men and had provided data for at least one measure, and were included in the analyses. Thirty-nine of these had provided answers for measures of PE, but not SIS/SES, giving a covariance coverage of 98.7%.

Symptoms of premature ejaculation were measured using the Checklist for Early Ejaculation Symptoms (CHEES; [14]), which is a five-item self-report questionnaire covering subjectively estimated ejaculation latency time, propensity to ejaculate with little stimulation, feelings of control over ejaculation, and feelings of frustration and relationship difficulties due to short ejaculation latency. The internal reliability was acceptable ($\alpha = 0.76$),

considering that the items, while all diagnostically relevant, are not necessarily highly correlated (e.g., some men have comparably long ELTs but are still frustrated by perceiving them as shorter than they wish). CHEES has been validated against stopwatch measured ejaculation latency time and other self-report measures. Validation studies have shown that it performs excellently in discriminating diagnosed patients from population-based controls (AUC = 0.98, 95% CI [0.97, 0.98]) [14].

Propensities for sexual excitation and inhibition were measured using the Sexual Inhibition/Sexual Excitation Scales—Short Form [15], which consists of 14 items that are answered on a Likert scale from 1 (strongly disagree) to 4 (strongly agree). The original study found that the measure represents three factors. The SES factor represents propensity to become sexually aroused in different intrapersonal (e.g., fantasizing about sex) or interpersonal (e.g., talking to someone on the phone who has a sexy voice) situations. The SIS1 scale consists of four items relating to how concern about sexual function or being distracted may reduce or inhibit sexual arousal. The SIS2 also consists of four items and is related to potential negative consequences of sexual activities (e.g., being caught in the act by others, catching a sexually transmitted disease). Internal reliability coefficients were not reported in the original study of the short-form questionnaire. In the present study, reliability coefficients were acceptable for SES and SIS1, but somewhat low for SIS2 ($\alpha_{SES}$ = 0.74, $\alpha_{SIS1}$ = 0.70, $\alpha_{SIS2}$ = 0.64). In a previous German validation study the SIS scales were found to have insufficient internal consistencies ($\alpha_{SES}$ = 0.75, $\alpha_{SIS1}$ = 0.56, $\alpha_{SIS2}$ = 0.67) [16]. The SIS/SES-SF has shown transcultural validity in French-Canadian [17] and German samples [16]. However, the three-factor structure has not always been replicated. Velten et al. [5] found support for a four-factor solution, where the SES scale was split into an intrapersonal and an interpersonal factor.

The survey was presented in Finnish. Items were translated from English into Finnish by a native Finnish speaker fluent in English, and back-translated to English by another native Finnish speaker fluent in English. The back-translated version was compared to the original version by a native English speaker, and any discrepancies were revised as appropriate.

The statistical analyses were conducted in Mplus version 7.4 for Mac [18]. There were no missing data for the PE measure, while 39 participants did not respond to any of the SES/SIS-SF items. All participants were included in the analyses. Structural equation modeling was used to estimate the associations between factors of interest, regressing premature ejaculation on the SIS/SES factors. Due to the SES/SIS-SF items being ordinal, a WLSMV (weighted-least-squares mean and variance adjusted) estimator was used [19,20]. The non-independence of observation due to some participants belonging to the same families was accounted for using the type = complex command. The $\chi^2$ test was not used to assess model fit due to the large sample size and subsequent risk of rejecting a true null hypothesis [21]. We instead used the following cutoffs for approximate fit indices to indicate an acceptable model fit: root mean square error of approximation (RMSEA) < 0.05, comparative fit index (CFI) $\geq$ 0.90, Tucker-Lewis index (TLI) $\geq$ 0.90. In case of poor model fit, we inspected modification indices to identify problems in the model. Exploratory factor analysis, where all items load on all factors, was used to explore alternative factor structures in the data.

## 3. Results

Descriptive statistics are found in Table 1. The structural equation model showed that sexual excitation was the strongest predictor of symptoms of PE ($\beta$ = 0.149, $p < 0.001$; Table 2), followed by the second sexual inhibition scale ($\beta$ = 0.091, $p$ = 0.005). Approximate fit indices were not satisfactory. An inspection of modification indices indicated that there was covariation between items within the SES scale that was not accounted for by the common factor. We therefore ran an exploratory structural equation model with four factors to analyze whether the SES items loaded on two separate factors.

**Table 1.** Descriptive statistics.

| Variable | | M | SD |
|---|---|---|---|
| Age | | 30.1 | 8.1 |
| SES | | 15.3 | 3.0 |
| SIS1 | | 8.2 | 2.9 |
| SIS2 | | 10.5 | 2.4 |
| | | n | % |
| Sexual orientation | Heterosexual | 2736 | 91.4 |
| | Homosexual | 123 | 4.1 |
| | Bisexual | 109 | 3.6 |
| | Other | 24 | 0.8 |
| Diagnostic categorization according to CHEES | Not indicative of PE | 2721 | 90.9 |
| | Indicative of PE | 218 | 7.3 |
| | Strongly indicative of PE | 53 | 1.8 |
| Subjectively estimated ejaculation latency time | I ususally don't ejaculate | 296 | 9.9 |
| | More than 10 min | 691 | 23.1 |
| | Between 5 and 10 min | 1094 | 36.6 |
| | Between 2 and 5 min | 675 | 22.6 |
| | Between 1 and 2 min | 181 | 6.0 |
| | Less than one minute | 55 | 1.8 |

Note. SES = sexual excitation scale; SIS1 = sexual inhibition subscale 1; SIS2 = sexual inhibition subscale 2. Means and standard deviations are calculated using sum scores.

**Table 2.** Parameter estimates for a structural equation model where premature ejaculation was regressed on the original three factors of sexual excitation and inhibition.

| Predictor | Unstandardized | | | | | Standardized | | | | |
|---|---|---|---|---|---|---|---|---|---|---|
| | b | SE | 95% CI | | $p$ | β | SE | 95% CI | | $p$ |
| | | | LL | UL | | | | LL | UL | |
| SES | 0.133 | 0.022 | 0.097 | 0.169 | <0.001 | 0.149 | 0.024 | 0.110 | 0.189 | <0.001 |
| SIS1 | −0.012 | 0.029 | −0.060 | 0.036 | 0.680 | −0.013 | 0.031 | −0.064 | 0.038 | 0.680 |
| SIS2 | 0.091 | 0.033 | 0.038 | 0.145 | 0.005 | 0.091 | 0.032 | 0.038 | 0.144 | 0.005 |

Note. Fit indices: $\chi^2(146) = 1676.2$, $p < 0.001$; root mean square error of approximation (RMSEA (90% CI)) = 0.059 (0.057, 0.062); comparative fit index (CFI) = 0.906; Tucker-Lewis index (TLI) = 0.890; R-square of dependent variable = 2.4%. CI = confidence interval; LL = lower limit; UL = upper limit; SES = sexual excitation scale; SIS1 = sexual inhibition subscale 1; SIS2 = sexual inhibition subscale 2.

The exploratory analysis found four factors with a clear pattern of loadings, with only item 8 loading on both SES factors. Fit indices were acceptable (RMSEA (90% C.I.) = 0.045 (0.040, 0.050); CFI = 0.983; TLI = 0.961; Table 3). The excitation factors were strongly positively correlated ($r = 0.577$), as were the inhibition factors ($r = 0.483$; Table 4). The excitation factors were weakly, mostly negatively, associated with the inhibition factors ($r$s range −0.206–145).

Next, we ran an SEM with four SES/SIS factors predicting PE (Table 5). Fit indices were acceptable. The results from the four-factor model make clear that symptoms of PE are associated with intrapersonal propensity for sexual excitation rather than a heightened sexual excitation in response to interpersonal sexual situations. SIS2 also predicted PE, but while it was statistically significant, the effect size was very small.

**Table 3.** Standardized loadings from an exploratory structural equation model with four factors.

| # | Original Item | SES1 | SES2 | SIS1 | SIS2 |
|---|---|---|---|---|---|
| 1 | When a sexually attractive stranger accidentally touches me, I easily become aroused. | **0.729 \*\*\*** | −0.008 | 0.026 | −0.026 |
| 3 | When I talk to someone on the telephone who has a sexy voice, I become sexually aroused. | **0.651 \*\*\*** | −0.041 * | 0.040 * | −0.050 * |
| 14 | When an attractive person flirts with me, I easily become sexually aroused. | **0.680 \*\*\*** | 0.131 \*\*\* | −0.080 \*\*\* | 0.021 |
| 8 | When I think of a very attractive person, I easily become sexually aroused. | **0.408 \*\*\*** | **0.427 \*\*\*** | 0.001 | 0.043 \*\* |
| 10 | When I start fantasizing about sex, I quickly become sexually aroused. | −0.030\*\* | **0.918 \*\*\*** | −0.045 \*\* | 0.002 |
| 11 | When I see other's engaged in sexual activities, I feel like having sex myself. | 0.130 \*\* | **0.502 \*\*\*** | 0.084 \*\*\* | −0.194 \*\*\* |
| 4 | I cannot get aroused unless I focus exclusively on sexual stimulation. | 0.023 | −0.047 * | **0.499 \*\*\*** | 0.188 \*\*\* |
| 9 | Once I have an erection, I want to start intercourse right away before I lose my erection/Once I am sexually aroused, I want to start intercourse right away before I lose my arousal. | 0.104 \*\*\* | 0.009 | **0.548 \*\*\*** | −0.041 |
| 12 | When I have a distracting thought, I easily lose my erection/my arousal. | −0.059 \*\* | 0.047 \*\* | **0.873 \*\*\*** | −0.014 |
| 13 | If I am distracted by hearing music, television, or a conversation, I am unlikely to stay aroused. | 0.000 | −0.042 * | **0.672 \*\*\*** | 0.105 \*\*\* |
| 2 | If I am having sex in a secluded, outdoor place and I think someone is nearby, I am not likely to get very aroused. | −0.048 * | −0.024 | -0.002 | **0.573 \*\*\*** |
| 5 | If I am masturbating on my own and I realize that someone is likely to come into the room at any moment, I will lose my erection/my sexual arousal. | 0.061 \*\* | 0.024 | 0.159 \*\*\* | **0.531 \*\*\*** |
| 6 | If I realize there is a risk of catching a sexually transmitted disease, I am unlikely to stay sexually aroused. | −0.022 | 0.078 * | 0.021 | **0.458 \*\*\*** |
| 7 | If I can be seen by others while having sex, I am unlikely to stay sexually aroused. | −0.002 | −0.019 | 0.005 | **0.821 \*\*\*** |

Note. # = item number in original scale. * $p < 0.05$ ** $p < 0.01$ *** $p < 0.001$. Bolded items load on their respective factors in the subsequent analyses (e.g., item #8 loads on SES1 and SES2).

**Table 4.** Estimated covariance and correlation matrices for the latent factors.

| Factor | SES1 | SES2 | SIS1 | SIS2 | PE |
|---|---|---|---|---|---|
| SES1 | 0.512 \*\*\* | 0.577 \*\*\* | 0.145 \*\*\* | −0.162 \*\*\* | 0.095 \*\*\* |
| SES2 | 0.161 \*\*\* | 0.152 \*\*\* | −0.100 \*\*\* | −0.206 \*\*\* | 0.144 \*\*\* |
| SIS1 | 0.064 \*\*\* | −0.024 \*\*\* | 0.384 \*\*\* | 0.483 \*\*\* | 0.038 |
| SIS2 | −0.067 \*\*\* | −0.047 \*\*\* | 0.174 \*\*\* | 0.337 \*\*\* | 0.054 * |
| PE | 0.040 \*\*\* | 0.033 \*\*\* | 0.014 | 0.018 * | 0.342 \*\*\* |

Note. Correlations (covariances) are presented above (below) the diagonal. Variances are found on the diagonal. * $p < 0.05$, *** $p < 0.001$, $n$ = 2953–2992.

**Table 5.** Parameter estimates for a structural equation model where premature ejaculation was regressed on four factors of sexual excitation and inhibition.

| Predictor | Unstandardized | | | | | Standardized | | | | |
|---|---|---|---|---|---|---|---|---|---|---|
| | b | SE | 95% CI | | *p* | β | SE | 95% CI | | *p* |
| | | | LL | UL | | | | LL | UL | |
| SES1 | 0.016 | 0.032 | −0.036 | 0.069 | 0.605 | 0.020 | 0.039 | −0.044 | 0.084 | 0.605 |
| SES2 | 0.226 | 0.059 | 0.128 | 0.323 | <0.001 | 0.151 | 0.038 | 0.088 | 0.214 | <0.001 |
| SIS1 | 0.009 | 0.032 | −0.043 | 0.031 | 0.780 | 0.009 | 0.034 | −0.046 | 0.065 | 0.780 |
| SIS2 | 0.084 | 0.033 | 0.030 | 0.138 | 0.010 | 0.084 | 0.032 | 0.030 | 0.137 | 0.010 |

Note. SES item 8 loads on both SES1 and SES2. Fit indices: $\chi^2(141) = 1170.0$, $p < 0.001$; Root mean square error of approximation (RMSEA (90% CI)) = 0.049 (0.047, 0.052); comparative fit index (CFI) = 0.937; Tucker-Lewis index (TLI) = 0.924. R-square of dependent variable = 2.8%. CI = confidence interval; LL = lower limit; UL = upper limit.

## 4. Discussion

In the present study, we analyzed the associations between sexual excitation, sexual inhibition, and symptoms of premature ejaculation in a large, population-based sample of Finnish men. The results indicated that men who suffer from PE also had an increased propensity for sexual excitation, with internal and external sexual stimuli eliciting stronger and more rapid sexual responses in these individuals. Consequently, it is plausible that a lower degree of stimulation triggers the ejaculation reflex (and in a shorter time) in men with elevated levels of PE symptoms compared to men with a lower propensity for sexual excitation. Furthermore, men with PE will probably have a higher degree of sexual excitation earlier in the sexual encounter, for example, so that they are close to the "point of no return" already at intromission. This gives limited opportunities to regulate the level of excitation during sexual activity, which may lead to ejaculating earlier than desired.

A clinical implication of the association between sexual excitation and PE is that psychoeducation and psychotherapeutic interventions can be useful [3,9]. The clinician can provide information regarding how the level of sexual excitation relates to the ejaculatory reflex and that people differ in their excitability. Further, they can help the patient to monitor their level of sexual excitement by working on identifying the idiosyncratic physiological and mental signals of sexual excitation. In the next step, they can work on identifying factors that affect sexual excitation (e.g., sexual behaviors, thought patterns, breathing, muscle tension), with the ultimate goal of learning to regulate the level of sexual excitement, and thus the timing of ejaculation. In addition, it appears plausible that high sexual excitation could respond to desensitizing interventions, which also seems to be the case [9,22]. These interventions could be further tested in future studies. This model of sexual excitement can also be relevant in non-clinical settings, such as sexual education and other forms of health promotion.

An interesting finding was that PE was not associated with sexual excitation stemming from social interaction, which is arguably inconsistent with the idea that performance anxiety would be causal of PE. Likewise, there was no association between PE and the SIS1 factor, which is thought to be related to performance anxiety. This is surprising given that precisely, performance anxiety is often mentioned as a likely culprit in PE etiology, however, very few studies have actually empirically investigated whether a statistical association between anxiety and PE exists. One of the few studies to have done so is a study by Ventus and colleagues [23], who in a longitudinal sample found a small but significant positive correlation between anxiety and PE within time, but found no evidence of anxiety predicting PE symptoms (or PE symptoms predicting anxiety) in a cross-lagged SEM model. In any case, the items of the SIS1 factor are centered around fear of losing erection or arousal, which is more closely related to erectile dysfunction than premature ejaculation. Indeed, the association between SIS1 and erectile dysfunction seems robust [10,24]. Nonetheless, the lack of association between PE and sexual inhibition could be construed as PE being more a problem of excessive excitation rather than lacking inhibition.

In the present study, we found no clear support for the idea that PE would be associated with sexual inhibition. This contradicts previous research by Nowosielski et al. [11], who found a small but significant association between SIS1 and lifetime prevalence of PE. It is possible, however, that this discrepancy may be due to differences between the studies in terms of defining PE. Depending on the definition, the prevalence of PE can vary from 1–2% (if the most stringent diagnostic criteria based on short ejaculation latency time are used) to 75% (if measures of subjective experience are used, see, e.g., [25]. Nowosielski et al. used a single question inquiring about self-estimated ejaculation latency time expressed in minutes, which does not cover other aspects of PE that are diagnostically relevant, whereas, in the present study, we used a composite score that measured (in addition to ejaculatory latency) diagnostically important aspects such as distress and perceived control over the ejaculatory reflex.

The present study should be interpreted bearing in mind the following limitations. First of all, the response rate of the present study was relatively low at 29%. However,

this is generally in line with response rates reported from other population-based studies inquiring about sensitive topics such as sexuality both nationally [26] (e.g., Haavio-Mannila and Kontula, 2003) and internationally [27,28]. Furthermore, the present sample appears to be comparable with other representative population-based samples in terms of several sexuality-related characteristics, such as mean age of first intercourse [29]. Secondly, we did not have at our disposal a clinical sample consisting of diagnosed PE patients (although it is more than plausible that many individuals in the present study would have, or at least be eligible for, a PE diagnosis). Having such as sample would have allowed for more clear-cut analyses of the associations between sexual inhibition/excitation and PE symptoms. Thirdly, we did not have experimental or longitudinal data at our disposal that would have allowed for testing causal relationships between variables measuring sexual excitation/inhibition and PE symptoms. To be able to investigate causal relationships between these would be an important contribution to the literature and help further our understanding of the etiology of PE. Finally, the CHEES measure used did not allow us to differentiate between different subtypes of PE. It would be useful for future studies to investigate how excitability is connected with PE in the different subtypes.

## 5. Conclusions

Symptoms of PE were associated with a greater propensity for sexual excitation. Importantly, this excitation was intrapersonal, as opposed to stemming from social activities. The results imply that men with PE may have stronger and more rapid reactions to sexual stimuli, which in turn may lead to ejaculating earlier than desired.

**Author Contributions:** Conceptualization, P.J.; formal analysis, D.V.; investigation, P.J.; resources, P.J.; data curation, D.V.; writing—original draft preparation, D.V. and P.J.; writing—review and editing, P.J. and D.V.; funding acquisition, P.J. All authors have read and agreed to the published version of the manuscript.

**Funding:** This research was funded by the Academy of Finland, grants number 319403 and 274521 (awarded to P.J.).

**Institutional Review Board Statement:** The study was conducted according to the guidelines of the Declaration of Helsinki and approved by the Ethics review board of Åbo Akademi University in Turku, Finland.

**Informed Consent Statement:** Informed consent was obtained from all subjects involved in the study.

**Data Availability Statement:** All raw data, as well as Mplus scripts are freely available here: osf.io/g8p6k.

**Conflicts of Interest:** The authors declare no conflict of interest. The funders had no role in the design of the study; in the collection, analyses, or interpretation of data; in the writing of the manuscript, or in the decision to publish the results.

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
