# Peer review of "Premature Ejaculation Symptoms Are Associated with Sexual Excitability: Empirical Support for the Hyperarousability Model"

_sexes, doi:10.3390/sexes2030027_

Round 1

Reviewer 1 Report

The manuscript entitled " Premature ejaculation symptoms are associated with sexual excitability: Empirical support for the hyperarousability model" from Ventus & Jern investigated the association of hyperexcitation and premature ejaculation symptoms. The results are interesting and present experimental data to strengthen the "consensus" that hyperexcitation status is a contributing factor to ejaculate prematurely and reinforce the importance of psychotherapy in cases of acquired/variable/subjective premature ejaculation. Overall, I think the manuscript would benefit from a discussion about the impact of the presented results on specific types of premature ejaculation, i.e. acquired/variable/subjective premature ejaculation as these types of premature ejaculation represent 82.3% of the studied population (manuscript Table 1). In addition, a discussion about the importance of psychotherapy on the basis of the presented data is warranted. No further issues were found.

Author Response

We would like to thank the reviewer for their useful comments.

Unfortunately, the measure utilized in the present manuscript does not allow us to differentiate between subtypes of PE. We now mention this on lines 253-255, and that future studies should investigate this interesting aspect.

We expanded the discussion on psychotherapeutic interventions on lines 197-209.

Changes in the manuscript are also marked with yellow color.

Reviewer 2 Report

It has been my pleasure to review this manuscript dealing with premature ejaculation.

Premature ejaculation is a fairly common problem in men and affects a significant part of them. It is necessary to study this subject in detail in order to improve the sexual life of people who have premature ejaculation problems.

This research attempts to demonstrate the possible associations between sexual arousal / inhibition and symptoms of premature ejaculation.

In general, the introduction seems correct to me, it provided enough information to put us in the context of the problem.

In the materials and methods section, the selection of the sample seemed a bit complex to me, but that is how it was selected and this is how the authors explain it.

The methodology developed also seemed adequate to me.
However, with the sole purpose of improving the quality of the manuscript, I would like to comment on some aspects that seem relevant to me:

In the materials and methods section (lines 99 onwards) it is explained that to measure the propensity for sexual arousal and inhibition, the Sexual Inhibition / Arousal Scale in its abbreviated form developed by Carpenter and his team was used. I think it would be necessary to explain in more detail the reliability of this scale by expressing at least the original Crombach alpha, and then verify the reliability in this specific sample, to be able to corroborate through the crombach alpha if it really exists. a good internal consistency in the scale that really measures what we intend to measure in this population.

On the other hand, after the discussion, it is necessary to create another section called conclusions.

Conclusions should be provided to support the results and should contain a summary of the most important findings of this research.

Finally, I have detected that in line 44 there is a bibliographic reference that is not well referenced. It is necessary to follow the same style throughout the manuscript. In this case, the style used was Vancouver.

Thanks

Kind regards

Author Response

We would like to thank the reviewer for their useful comments.

We expanded the section on reliability of the measure on lines 113-117. Unfortunately, no internal coeefficients were reported in the original study of the short-form measure, however, compared to a German validation study, the reliability was higher in the present sample. As for reliability, by using latent modelling in the present analyses, we are able to account for measurement error. Further, the exploratory factor analysis did indeed indicate that the SES scale was actually measuring two different aspects of excitation.

We have corrected the reference on line 44.

We added conclusions as a separate section on lines 256-260.

Changes in the manuscript are also marked with yellow color.

Reviewer 3 Report

This paper is important in understanding how premature ejaculation is associated with sexual excitation and inhibition and could be used in tailored interventions for men's sexual health and wellbeing.

Some additional reviewer feedback are provided below:

-It would be great to have some operational definition of sexual inhibition and sexual excitation and how this definition has been related to the model development process.

-It would be great to know about the reliability/validity of the CHEES (e.g. for internal consistency)

-line 216-220; It would be great to have some proposed interventions in your findings in the discussion about the importance of using composite score and how the clinical or other psychological interventions would be helpful for future studies.

-It would have been interesting to learn more about the demographic sample, possibly provide a line in explaining urban/rural setting etc.

-line 201-205; It was important to bring in the piece of health literacy and health promotion in the discussion section. It would provide a better readability on the importance of sexual education/ sexual health counselling can improving sexual health.

Author Response

We would like to thank the reviewer for their useful comments.

  1. We clarified the meaning of the concepts on lines 34-38.
  2. We expanded the description of reliability on lines 97-103.
  3. We expanded the discussion on clinical interventions that could be used in future studies on lines 197-209.
  4. Unfortunately, no further demographic data were gathered. The sampling frame is population-based, and thus includes people from every part of the country, of all levels of socioeconomic status. We elaborated on the population registry on lines 79-81.
  5. We expanded the discussion on sexual education on lines 197-209.

Changes in the manuscript are also marked with yellow color.

Round 2

Reviewer 2 Report

It was my pleasure to review this second improved version of the manuscript.
The authors have endeavored to respond to and apply the reviewers' recommendations, notably improving the quality of the manuscript.
The manuscript seems correct to me for its publication in this Journal and I have no objection to it.
I think a good job was done.

Thank you

Kind regards